# Integrated Multiomics Analysis of Salivary Exosomes to Identify Biomarkers Associated with Changes in Mood States and Fatigue

**DOI:** 10.3390/ijms23095257

**Published:** 2022-05-09

**Authors:** Whitaker Cohn, Chunni Zhu, Jesus Campagna, Tina Bilousova, Patricia Spilman, Bruce Teter, Feng Li, Rong Guo, David Elashoff, Greg M. Cole, Alon Avidan, Kym Francis Faull, Julian Whitelegge, David T. W. Wong, Varghese John

**Affiliations:** 1The Drug Discovery Lab, Department of Neurology, David Geffen School of Medicine, 710 Westwood Plaza, University of California Los Angeles, Los Angeles, CA 90095, USA; wcohn@mednet.ucla.edu (W.C.); chunnizhu@mednet.ucla.edu (C.Z.); jcampagna@mednet.ucla.edu (J.C.); tbilousova@mednet.ucla.edu (T.B.); pspilman@mednet.ucla.edu (P.S.); bteter@ucla.edu (B.T.); 2Center for Oral/Head & Neck Oncology Research, Laboratory of Salivary Diagnostics, School of Dentistry, 10833 Le Conte Avenue, University of California Los Angeles, Los Angeles, CA 90095, USA; fli@dentistry.ucla.edu (F.L.); dwong@dentistry.ucla.edu (D.T.W.W.); 3Department of Medicine Statistics Core, David Geffen School of Medicine, 1100 Glendon Avenue, University of California Los Angeles, Los Angeles, CA 90095, USA; rongguo@mednet.ucla.edu (R.G.); delashoff@mednet.ucla.edu (D.E.); 4Department of Neurology, David Geffen School of Medicine, 710 Westwood Plaza, University of California Los Angeles, Los Angeles, CA 90095, USA; gregorycole@mednet.ucla.edu (G.M.C.); avidan@mednet.ucla.edu (A.A.); 5Pasarow Mass Spectrometry Laboratory, Jane and Terry Semel Institute for Neuroscience and Human Behavior, David Geffen School of Medicine, 760 Westwood Plaza, University of California Los Angeles, Los Angeles, CA 90095, USA; kfaull@mednet.ucla.edu (K.F.F.); jwhitelegge@mednet.ucla.edu (J.W.)

**Keywords:** fatigue, cognitive impairment, exosomes, biomarkers, saliva, multiomics, proteomics, microRNA

## Abstract

Fatigue and other deleterious mood alterations resulting from prolonged efforts such as a long work shift can lead to a decrease in vigilance and cognitive performance, increasing the likelihood of errors during the execution of attention-demanding activities such as piloting an aircraft or performing medical procedures. Thus, a method to rapidly and objectively assess the risk for such cognitive fatigue would be of value. The objective of the study was the identification in saliva-borne exosomes of molecular signals associated with changes in mood and fatigue that may increase the risk of reduced cognitive performance. Using integrated multiomics analysis of exosomes from the saliva of medical residents before and after a 12 h work shift, we observed changes in the abundances of several proteins and miRNAs that were associated with various mood states, and specifically fatigue, as determined by a Profile of Mood States questionnaire. The findings herein point to a promising protein biomarker, phosphoglycerate kinase 1 (PGK1), that was associated with fatigue and displayed changes in abundance in saliva, and we suggest a possible biological mechanism whereby the expression of the PGK1 gene is regulated by miR3185 in response to fatigue. Overall, these data suggest that multiomics analysis of salivary exosomes has merit for identifying novel biomarkers associated with changes in mood states and fatigue. The promising biomarker protein presents an opportunity for the development of a rapid saliva-based test for the assessment of these changes.

## 1. Introduction

The onset of fatigue confers an increased risk of impaired cognitive performance, resulting in what is referred to as cognitive fatigue (CF) [1,2]. This is characterized by an inability to maintain optimal performance during prolonged cognitive effort [3]; it can manifest after long-duration cognitive activity, sleep deprivation [4], or excessive exercise [5]. CF is of particular concern for those involved in attention-demanding occupations such as surgeons performing extended operations and pilots navigating long flights and may result in catastrophic accidents. Typically, the extent of fatigue and thus the risk of CF is determined by self-reported assessments, which are subjective and unreliable. Thus, a method for objectively assessing physiological alterations that are associated with an increased risk of CF would be of value, and the identification of biomarkers associated with fatigue has the potential to be the first step in the development of rapid tests to determine the risk of CF.

Fatigue-associated changes may be due, in part, to biochemical changes in cellular signaling processes and observed in the molecular composition of saliva. While saliva is obtained easily and noninvasively in sufficient quantities for analyses, the ability to identify fatigue biomarkers is hindered by the complexity and concentration dynamic range of the salivary proteome and transcriptome. Here, we have attempted to overcome this challenge by assessing biomarkers in exosomes, a subset of extracellular vesicles (EVs), in saliva.

Exosomes are 50–100 nm diameter [6] particles of endosomal origin that occur in body fluids and the extracellular space of tissues. They are comprised of a lipid bilayer that carries cargo in their interior and is attached to their surface. Like other exosomes, the surface of salivary exosomes is enriched in tetraspanin marker proteins CD9, CD63, and CD81 [7,8], important for the organization of membrane domains. Exosomes also contain tumor susceptibility gene 101 (Tsg101) and ALG-2-interacting protein X (ALIX) that are part of the Endosomal Sorting Complexes Required for Transport (ESCRT) machinery involved in intracellular vesicle formation and sorting of cargo [6].

In a process still being elucidated, during the formation, exosomes are loaded with cytoplasmic proteins, nucleic acids, including microRNAs, and membrane constituents that reflect the biochemistry and status of the parent cell at the time of the exosome biogenesis [9]. Exosomes are released from all cell types, including neurons, and can act as intercellular signal carriers. Sleep deprivation is associated with a decline in cognitive function and alterations in levels of intracellular proteins and nucleic acids [10,11], as well as circulating signaling molecules [12]. Thus, exosomes derived from individuals experiencing sleep deprivation may carry molecular signals reflective of changes in the physiology of the central nervous system (CNS) associated with the onset of fatigue-associated cognitive impairment [13].

We hypothesize that the potential for salivary exosomes to reflect changes in cognitive function is due to the presence of anatomical connections from the CNS, specifically innervation of Cranial Nerves VII and IX from the superior and inferior salivary nuclei to the oral cavity and the parotid and submandibular glands [14], or directly from the blood through the oral cavity vasculature. The hypothesis that salivary exosomes carry signals that influence and are indicative of the changes in brain function is supported by reports on salivary exosome proteins changing with HIV-associated cognitive deficits, in traumatic brain injury concussion-related cognitive fatigue, and in monozygotic twins discordant for chronic fatigue syndrome [15,16,17]. Further, proteins such as beta-amyloid [18] and tau [19], implicated in the impaired cognitive function associated with Alzheimer’s disease, change in abundance in the total saliva protein pool and are present in bloodborne brain-derived EVs [20]. These findings make it reasonable to postulate that other proteins and biomarkers related to cognitive function may be contained in salivary exosomes.

Autonomic nervous system alterations also play a role in the manifestation of fatigue [21] and saliva production [22]; therefore, it is also likely that exosomes released by neurons of the autonomic nervous system are present in salivary exosomes.

To test our hypothesis that exosomes originating from neuronal cells are present in saliva and carry biomarkers reflecting physiological states of fatigue, saliva from medical and dental resident participants was collected before and after a 12 h work shift. Participants were asked to complete a modified Profile of Mood States (PoMS) questionnaire both before and after their work shift, and the changes in subscale scores for self-assessed mood states such as fatigue were used to generate Total Mood Disturbance (TMD) scores. The PoMS method of assessment was chosen because, in Fogt et al. [2], increased fatigue, as reflected by the PoMS score, was directly correlated with decreased cognitive performance as determined by the Stroop Color-Word Conflict Test. 

Exosomes isolated from the collected saliva were subjected to multiomics analyses, including quantitative proteomics and miR-omics, to identify proteins and microRNAs associated with changes in TMD scores.

## 2. Results

### PoMS Analysis and Assignment to Study Groups

Participant answers on the PoMS questionnaire were used to calculate pre- and post-work shift TMD scores. Each of the mood state subscales—TA, DD, AH, FI, CB, and VA (Figure 1A)—contributed to the TMD score (Figure 1B and Appendix A, Appendix A), with only the VA being subtracted from the total of the others because increased vigor and activity are associated with an improved, rather than worsening, mood. Therefore, an increase in TMD indicated a decline in mood states post shift. Significantly increased scores were observed for CB and FI and a decreased score for VA (Appendix A and Figure 1A). The TMD score also significantly increased from pre-shift (53.07 ± 20.21) to post shift (65.99 ± 24.83) (*p* < 0.05), indicating that the mood of most participants worsened during the shift (Figure 1B). Not all participants recorded a positive TMD. A decrease in TMD score was observed for 12 of the 36 (33%) participants, indicative of no change or an elevated mood post shift (Figure 1C).

As described in *Methods* Section, the TMD score was subsequently used to segregate saliva samples into three groups: Test, Discovery, and Validation (Figure 1C).

The purpose of the Test group was to establish the validity of analytical methods and testing of potential biomarker candidates using six participants with a negative TMD difference whose saliva would not be predicted to harbor biomarkers associated with deteriorating mood. The Discovery group focused on participants with self-reported increased fatigue (Figure 1D) and a decline in mood and consisted of the 20 individuals with the largest increase in TMD. The Discovery group would also be expected to exhibit physiological changes associated with fatigue and an increased risk for CF. The Validation group included participants with only slightly positive and negative changes in TMD and FI, whose saliva samples underwent proteomics and qPCR analysis. A narrower group of proteins and miRNAs identified from the larger Discovery group analysis were measured in the Validation group, and any correlations to PoMS subscales were determined.

## 3. Multiomics Analysis of Test Group Salivary Exosomes

### 3.1. Quantitative Global Proteomics on Salivary Exosomes

Bottom-up proteomics analysis identified a total of 118 unique proteins in exosomes enriched from the saliva of Test group participants. Reporter ion-based quantification of proteins using Tandem Mass Tags (TMT) revealed quantifiable differences in the abundance of 98 proteins extracted from the salivary exosomes of residents before and after 12 h work shifts (Figure 2A). While changes in the abundance of only two proteins, BPI fold-containing family A member 2 (BPIFA2; fold change = 1.91, *p* = 0.02) and small proline-rich protein 3 (SPRR3; fold change = −2.58, *p* = 0.05) were statistically significant, many other proteins showed trends and were close to reaching significance (*p* < 0.05; Appendix A). GO classifications revealed an abundance of exosome-associated proteins (Appendix A), including four altered membrane-bound proteins (Appendix A).

### 3.2. Verification of Protein Measurements in Neuron-Derived Exosomes by Targeted MS

In the single Test group of participants from which neuron-derived exosomes were isolated, as many as 50 of the proteins identified from the immunoprecipitation using exosomal cell surface markers CD9, CD63, CD81 (Appendix A) were subsequently also identified using targeted MS for neuron-derived exosomes after isolation of salivary exosomes using neuron cell surface marker (CD171) (Figure 2B and Appendix A). The targeted MS was performed for proteins with abundant peptides that contained no post-translational modifications. Four proteins—BPIFA2, CSTB, PIGR, and PKM—in neuron-derived exosomes correlated well with global proteomics used for exosomes isolated using pan-exosomal markers, but the protein fold change observed in the targeted MS was higher compared with global proteomics (Figure 2C).

### 3.3. MicroRNA Analysis Using NanoString

The NanoString platform was used to determine the abundance of a panel of 800 biologically relevant miRNAs. While not all the miRNA species were quantifiable above background in salivary exosomes, the analysis revealed 22 miRNAs to be significantly changed between pre- and post-work shifts (absolute fold change ≥ 1.2, *p* < 0.05) (Figure 2D and Appendix A). Several of the significantly altered miRNA were found to target genes encoding proteins that were also determined to change in abundance between pre- and post-work shifts (Appendix A). An inverse relationship between some identified miRNAs and their associated protein was observed.

### 3.4. Verification of miRNA Measurements Using qPCR

NanoString abundance measurements were validated for two select miRNAs (miR1296-3p and miR519d-3p) using qPCR. Values of % change between pre- and post-work shift for the miRNAs in the 6 Test group participants show that measurements made using the NanoString platform were qualitatively verified by qPCR (Figure 2E). This result provided the reassurance of the reliability of NanoString miRNA abundance measurements needed for subsequent analysis of the Discovery group saliva samples. The raw qPCR results are presented in Appendix A.

## 4. Biomarker Identification in Discovery Group Saliva

### Identification of Additional Significantly Altered Protein and miRNA

Quantitative, bottom-up proteomics analyses on the salivary exosomes from the larger Discovery group (*n* = 20) resulted in the identification of an increased number of proteins when compared with the Test group. Among the 309 proteins quantified (Appendix A), the abundance of 7 of these was determined to be significantly altered between pre- and post- work shifts (absolute fold change ≥ 1.2 with a *p* value < 0.05), and 7 additional proteins displayed trends that were close to reaching statistical significance (Figure 3A). All of the proteins that exhibited a difference in absolute fold change ≥ 1.2 with a *p* value < 0.1 and their associated miRNA are displayed in Table 1.

NanoString miRNA analysis showed that 22 miRNAs significantly changed between pre- and post-work shifts (absolute fold change ≥ 1.2, *p* < 0.05, Appendix A, Figure 3A). In addition, 69 miRNAs changed in correlation with changes in either the PoMS TMD (32 miRNAs) or FI (37 miRNAs) subscale (Appendix A, respectively). As in the Test group, the abundance of several miRNAs, miR-3185, miR-642-5p, and miR-134-3p, was found to inversely correlate with the protein encoded by one of their target genes (Figure 3B and Table 1). This relationship highlights three potential proteins (Phosphoglycerate Kinase, gene PGK1; Polymeric Immunoglobulin Receptor, gene PIGR; and Tryptophan 5-Monooxygenase Activation Protein Zeta, gene YWHAZ) and miRNA (miR-3185, miR-642a-5p, miR-134-3p) biomarkers to be quantified in Validation group saliva. One of these miRNAs, miR-642a-5p, weakly correlated with changes in the PoMS FI subscale (Appendix A). Additional bioinformatic approaches, including gene set enrichment, GO classification and pathway analysis (Figure 3C), and functional protein network analysis (Figure 3D), were utilized to elucidate the potential biological roles of the identified proteins in increased TMD score.

In addition, correlations of NanoString and qPCR analyses were determined for miR642a, miR3185, and miR3185 (Appendix A).

Of the six proteins that were increased and the eight proteins that were decreased in the Discovery group (Table 1), three are encoded by genes that could be a target of three of the miRNAs that changed in abundance in the opposite direction to that of the protein (as expected for the normal downregulation of gene expression by a miRNA).

## 5. Confirmation of Biomarkers and Fold Change in Validation Group Salivary Exosomes

### 5.1. Validation of Protein Biomarkers

Proteomic analysis of salivary exosomes from Validation group participants confirmed the presence of 12 of the 14 candidate biomarkers identified in the Discovery group (Table 1, Figure 4A). As observed in the Discovery group, the mean fold change in the abundances of five proteins CSTB, DDP4, FABP5, PIGR, and YWAZ maintained an inverse relationship with changes in TMD score. We note, however, that none of these changes were statistically significant (*p* < 0.05). The abundance of six proteins identified in the Validation group (DPP4, BPIFA2, CA6, AMY1A, LEG1, and DMBT1) did not correlate with changes in TMD or FI score but exhibited mean fold changes similar to what was observed in the Discovery group. One protein, PGK1, identified in the Discovery group was also identified in the Validation group, and its fold change showed a positive but weak correlation with changes in FI score (Figure 4B).

### 5.2. Validation of miRNA Biomarkers

The Validation Group samples were assessed for a total of 13 miRNAs using qPCR. These miRNAs were chosen based on the Discovery group. NanoString results which showed either large or highly significant changes in their levels with either pre-shift/post shift or change with PoMS subscales TMD or FI (Appendix A), specifically 13 miRNAs, were miR3185, miR28, miR1296, miR182, miR614, miR4536, miR140, miR1257, miR518e, miR105, miR126, miR642a, and miR134. The abundance of one miRNA in the validation group—miR3185—was found to moderately correlate with PoMS FI subscale score differences (Figure 4C).

### 5.3. PGK1 Protein and miR3185 in Saliva as Potential Biomarkers of Fatigue

In the Test group participants’ salivary exosomes (with a lower rather than increased TMD score post shift), the fold change in PGK1 was negative, and in the Discovery group, with participants who recorded increased TMD and FI scores, it was positive (Figure 5A). In 15 of 30 participants in either the Discovery or Validation groups, wherein PGK1 was identified, there was an inverse correlation with miR3185 in both positive and negative abundance ratios (Figure 5B).

### 5.4. Integration of Test, Discovery, and Validation Data Reveals Proteins Associated with Work

When changes in protein abundances from the Test, Discovery, and Validation groups are combined, five proteins (AMY1A, BPIFA2, CA6, DPP4, and LEG1) significantly increase, and two proteins (SBSN and DMBT1) significantly decrease after a 12 h work shift (absolute fold change ≥ 1.2, *p* < 0.05) (Appendix A).

## 6. Discussion

Proteomics analysis of Test group saliva confirmed successful enrichment of exosome populations and identified changes in the abundance of several proteins pre- and post-work shift. Some of the proteins with altered abundances are known to be associated with Chronic Fatigue Syndrome (CFS), including alpha amylase 1 (AMY1A), cystatin-B (CSTB), polymeric immunoglobulin receptor (PIGR), deleted in malignant brain tumors 1 protein (DMBT1), lysozyme C (LYZ), and ras-related C3 botulinum toxin substrate 1 (RAC1) [17,26,27,28]. The abundance of these CFS-associated proteins increased in some instances and decreased in others without correlation to the ‘improved’ mood reported by the Test group (Appendix A). It is significant that two biomarkers, protein PGK1 and miR3185, show abundance in saliva exosomes and change directions when the TMD score difference (Figure 2C) switches from negative to positive (Figure 4B,C); this shows that these biomarkers are responsive to changes in mood state and fatigue.

A subset of proteins with altered abundance in the Test group are membrane-bound proteins (Appendix A), an appealing characteristic as these potential biomarkers could possibly be identified without requiring exosome lysis in future studies.

In a single Test group participant, four of the proteins (BPIFA2, CSTB, PIGR, and PKM) with altered abundance identified from pan-exosome-isolated samples were qualitatively validated using global proteomics/Proteome Discoverer in neuron-derived exosomes (Appendix A). A notable increase in fold change was observed when using a targeted mass spectrometry approach for the neuron-derived exosomes. This discrepancy most likely results from a systematic underestimation of quantitative ratios caused by co-fragmentation of undesirable peptides when using isobaric mass tags such as those used in the untargeted proteomic analysis [29]. This ratio compression does not occur when using targeted, label-free quantification strategies resulting in more pronounced fold changes. Taking this into account, smaller fold changes need to be considered significant when using isobaric mass tags for quantitative proteomics in the discovery group analysis, and proteins of interest should be further analyzed using targeted approaches. Alternatively, these augmented fold change results may also be due to enhanced protein responses in neuron-derived exosomes, which become reduced in magnitude when diluted in total exosomes.

miRNA analysis on the Test group using the NanoString platform identified 22 miRNAs as significantly changed between pre- and post-work shifts, 12 of which were also found to change in the Discovery and Validation groups. These measurements were subsequently validated for two miRNAs (miR1296-3p and miR519d-3p) using qPCR, providing reassurance of the reliability of the NanoString platform. Several of these miRNAs were also found to exhibit changes in abundance opposite to that of the identified protein encoded by their target genes, suggesting a mechanism of gene regulation that is influencing the abundances of identified proteins. Omics analyses in the Test group confirmed our ability to identify exosomal proteins and their associated miRNAs that are detectable and may be altered pre- and post-work shift.

Global proteomic analysis of the larger Discovery group samples identified a considerably greater number of significantly changed proteins, including increases in AMY1A, BPI fold-containing family A member (BPIFA2), dipeptidyl peptidase 4 (DPP4), and decreases in small proline-rich protein 3 (SPRR3), fatty acid-binding protein 5 (FABP5), 14-3-3 protein zeta/delta (YWHAZ), and DMBT1. Four proteins that were altered—AMY1A (increased), CSTB (decreased), PIGR (decreased), and DMBT1 (decreased)—are known to be associated with CFS. Several other proteins, including liver-enriched gene 1(LEG1), carbonic anhydrase 6 (CA6), suprabasin (SBSN), phosphoglycerate kinase 1 (PGK1), and cellular retinoic acid-binding protein 5, demonstrated close to significant changes in abundance and were also considered potential biomarkers. Gene set enrichment, GO classification and pathway analysis, and functional protein network analysis were utilized to help understand the potential biological roles of the identified proteins. These analyses highlighted the potential existence of regulated fatigue-associated protein networks that generate ATP in response to energy demand or cellular stress. PGK1 catalyzes the formation of ATP from ADP and 1,3-diphosphoglycerate, playing an important role in glycolysis and energy homeostasis [30]. AMY1A hydrolyzes 1,4-alpha-glucosidic bonds in oligosaccharides and polysaccharides, yielding glucose that is then available to generate ATP [31]. DPP4 also influences glucose levels by deactivating incretins, which normally stimulate the release of pancreatic insulin [32].

NanoString miRNA analysis on the Discovery group identified 69 miRNAs that significantly changed and correlated with PoMS TMD or FI (Appendix A). Interestingly, some of these miRNAs exhibit changes in other CNS pathologies. For example, hsa-miR-142-3p is increased in individuals who have experienced a mild traumatic brain injury [33]. We hypothesize that the expression of these miRNAs is sensitive to changes in cognitive function and that they regulate the expression of biologically relevant proteins and pathways. Integrated analysis of the two-omics datasets was used to determine if any significantly altered miRNAs were known to regulate target genes encoding significantly altered proteins. miRNA-protein/gene pairs were selected if the direction of change in the miRNA was in the opposite direction of the change in the protein, considering the typical mechanism of downregulation of a gene mRNA by upregulated miRNA. This analysis identified three miRNA-protein/gene pairs: miR-3185—PGK1, miR-642a—PIGR, and miR-134—YWHAZ.

Validation group saliva was used to determine if the candidate biomarkers identified in the Discovery group analysis correlated with TMD or FI. Proteomic analysis identified 12 of the 14 proteins altered in the Discovery group to be present in Validation group exosomes and, while not statistically significant, the mean fold change in CSTB, DDP4, FABP5, PIGR, and YWAZ maintained an inverse relationship with changes in TMD score. Additionally, PGK1 maintained and positive but weak correlation with changes in FI score. It should be noted that the magnitude of TMD and FI difference scores in the validation group was significantly smaller than those of the Discovery group, which may make biomarker validation more challenging and is most likely evidenced by a lack of statistical significance. When the magnitude of FI score differences is considered, PGK1 would be an interesting protein for continued evaluation as a biomarker of cognitive fatigue. When the Validation group and Discovery group data were combined, the abundance of the other six candidate biomarkers (DPP4, BPIFA2, CA6, AMY1A, LEG1, and DMBT1) were still significantly altered between pre- and post-work saliva but were not determined to be associated with either TMD or FI. This notable observation highlights that many of the originally identified potential biomarkers may be associated with biological processes altered by work alone that are not impacted by changes in TMD or FI score. Therefore, these proteins may still have value as biomarkers of biologically relevant phenomena unrelated to mood- and fatigue-associated cognitive impairment.

Of 13 miRNAs selected for continued evaluation in Validation group samples, the abundance of miR3185 was correlated with TMD and FI. miR3185 is of particular interest because it regulates the target gene PGK1, a protein identified as a potential biomarker correlated with fatigue [34]. The relation between miR3185 and PGK1 was strengthened by their inverse correlation, suggesting a potential biological mechanism for regulation of the PGK1 gene by miR3185 may be induced by fatigue. The inversely correlated levels of miR3185 and PGK1 could represent a coregulated set that is not only a biomarker of fatigue but could possibly contribute to a mechanism of fatigue induction or relief. The levels of miR3185 and PGK1 not only correlated with the degree of mood disturbance assessed by the POMS FI subscale, but the correlation extended beyond increased FI to decreased FI; among subjects whose PoMS FI difference was negative (reduced fatigue), miR3185 increased, and PGK1 decreased.

PGK1 deficiency is associated with anemia syndromes that include the progressive onset of weakness, fatigue, and lassitude [35] and motor neuron vulnerability in spinal muscular atrophy (SMA) [36]. The increase in PGK1 with increased mood disturbance may represent a compensatory response to boost energy levels, but this is speculative and might be elucidated by following PGK1 levels in salivary exosomes over time during a demanding work shift.

Little is known about miR3185, other than it is specific to primate genomes [37] and reported to be increased in cardiac tissues in cases of mechanical asphyxia [38] as well as associated with increased survival in liver cancer [39]. PGK1 and miR3185 are both attractive biomarker targets that could potentially be used to detect the onset of mood- and fatigue-associated cognitive impairment in salivary exosomes.

There were some important limitations to this exploratory study for saliva biomarkers. The PoMS questionnaire determines total mood disturbance, and not specifically CF, but two subscales—fatigue–inertia (FI) and confusion–bewilderment (CB)—that were significantly worsened in the Discovery group very likely affect cognitive performance, as previously reported for PoMS-determined fatigue [2]. Confirmation of this association will require further study and the use of objective CF-specific testing. We further note that work shifts were not controlled for time of day, that is, the impact of circadian rhythm on exosomal content or for the activity levels of each of the participants, which could affect their reported PoMS scores. We did not calculate effect sizes for comparison in the present study. Additionally, while the findings from neuron-derived exosomes suggest concordance/overlap with findings from global analysis of exosomes isolated without neuron-specific markers, these selected exosomes were from a single participant, and the findings require validation with greater *n* numbers. Similarly, the miRNA/proteinbiomarkers described require validation. Further, while some of the candidate biomarkers have interesting reported roles in fatigue or CFS, the analysis performed here does not establish a causal relationship between the PoMS TMD score/fatigue-inertia score and the changes in exosomal content pre- and post-shift.

In conclusion, our study identified proteins and miRNAs in salivary exosomes that correlate with changes in mood state and fatigue as measured by the PoMS questionnaire. They represent possible biomarkers that can be quantified using saliva with the potential to reveal an increased risk for loss of vigilance and decline in cognitive performance. These results add to the growing knowledge of detectable changes in the biomolecular composition of exosomes in various pathologies and point to a promising candidate biomarker, PGK1, in saliva, as well as suggest a possible mechanism in which expression of the PGK1 gene is regulated by miR3185 in response to changes in fatigue. This salivary biomarker requires further clinical validation in larger well-defined cohorts. The limitations of the current study were the small sample size and the potential inaccuracies associated with subjective self-assessment of mood states. Despite these limitations, this study demonstrates the value of using an integrated multiomics approach to the identification of novel mechanisms and biomarkers in salivary exosomes, with the possibility of developing a rapid saliva-based antigen test for cognitive fatigue.

## 7. Methods

### 7.1. Participants

Donors were recruited from UCLA medical and dental residents. A total of 36 residents participated. The research was approved by the UCLA IRB committee (UCLA IRB # 17-000317). Residents were given information about the research, and they gave oral consent for participation in this study. 

### 7.2. Whole Saliva Collection

Saliva samples were collected over a period of 60 min immediately before and following a 12 h work shift in 50 mL conical tubes. Samples were centrifuged (2600 rcf, 4 °C, 15 min), and the supernatants were aliquoted (1 mL) in microcentrifuge tubes containing Superase RNase inhibitor (1 µL, Thermo Fisher Scientific, Cat #AM2694, Waltham, MA, USA) and stored at −80 °C until processing. 

### 7.3. The Profile of Mood States (PoMS) Questionnaire

Mood states were accessed using a modified version of the PoMS questionnaire [40,41,42,43]. This consisted of a 62-item inventory of six subscales: tension–anxiety (TA), depression–dejection (DD), anger–hostility (AH), vigor–activity (VA), fatigue–inertia (FI), and confusion–bewilderment (CB). Responses were provided on a 5-point scale ranging from 1 (not at all) to 5 (extremely). The global indicator Total Mood Disturbance (TMD) is defined as: TMD = (AH + CB + DD + FI + TA) − VA. An increase in TMD suggests the onset of mood disturbances that would be considered unfavorable for optimum vigilance and cognitive performance, such as increased fatigue which has been associated with increased cognitive fatigue and decreased cognitive performance [2]; decreases in TMD reflect positive changes in mood, for example, a decrease in tension and anxiety. 

### 7.4. Separation of Saliva Samples by PoMS TMD Score

Based on TMD scores, subject saliva samples were assigned to 3 groups, the Test group, Discovery group, or Validation group (Figure 6), similarly to a previously reported approach used for the analysis of saliva samples for biomarkers of traumatic brain injury [16]. The Test group comprised saliva from 6 participants with a negative TMD difference, that is, those who reported no change or an improvement in mood as a result of the work shift. The Discovery group comprised 20 subjects with the greatest increase in TMD score post shift. Saliva samples from both groups underwent exosome isolation followed by proteomics and Nanostring miRNA analyses. The Validation group consisted of 10 subjects with nearly unchanged or slightly increased or decreased TMD post-shift scores. For the Validation group, the same methods were used as for the other two groups, but proteomics were targeted for select proteins, and qPCR was performed for select genes. These groups were established based on the hypothesis that potential biomarkers of fatigue ‘discovered’ in individuals reporting fatigue (Discovery group) could be tested for their potential as biomarkers in the group that reported an opposite ‘improved’ change in mood (Test group) based on the supposition that these biomarkers would either be unchanged or change in the opposite direction in the Test group. The potential biomarkers were again ‘validated’ in the group with a mix of scores (Validation group). 

### 7.5. EV Isolation and Enrichment for Exosomes

Salivary EVs were isolated using magnetic microsphere-based immunoprecipitation (IP) modified from established methods [44]. Frozen saliva aliquots were thawed at 37 °C, spiked with HALT Protease and Phosphatase Inhibitor Cocktail (Thermo Fisher Scientific, Cat # 78440, Waltham, MA, USA), and diluted threefold with ice-cold phosphate-buffered saline (PBS) and centrifuged (13,000 rcf, 20 min, 4 °C). Supernatants were then incubated (overnight, 4 °C) with a mixture of antibodies specific for various exosomal surface markers, including tetraspanins CD9, CD63, and CD81 (all Thermo Fisher Scientific, Cat # 10626D, 10628D, and 10630D, respectively, Waltham, MA, USA) that were previously desalted (Zeba™ Spin Desalting Columns, 7K MWCO, 0.5 mL, Thermo Fisher Scientific Cat # 89882, Waltham, MA, USA) and conjugated to Dynabeads (Invitrogen DYNAL Dynabeads M-270 Epoxy, Thermo Fisher Scientific Cat # 14301, Waltham, MA, USA) according to the manufacturer’s protocols. The isolated exosomes were used for bottom-up proteomics by mass spectroscopy (MS) and miRNA analysis.

For isolation of salivary exosomes originating from neurons (performed for a single participant in the Test Group), Dynabeads conjugated to antibodies specific for a neuronal surface marker CD171 (Thermo Fisher Scientific, Cat # MA5-14140, Waltham, MA, USA) was used. After incubation, the diluted saliva samples with Dynabeads were set on a magnetic bar for 1 min, after which supernatant was discarded. The beads destined for proteomics analysis were subsequently washed once with 1× PBS, twice with 0.15 M citrate phosphate buffer (pH 5.2), and once again with 1× PBS. For beads destined for miRNA analysis, 0.1% BSA was added to both 1× PBS washes. The isolated exosomes were used for targeted proteomics by MS.

### 7.6. Quantitative Global Proteomics Analysis

Preliminary protein quantification assays indicated that the amount of total protein in samples post immunoprecipitation-based EV enrichment from saliva was very low (<2 μg). Because of the limited amount of protein in each sample, protein levels were normalized after the proteomic analysis. After the proteins were digested with trypsin, peptides from each sample were chemically modified with different isobaric tandem mass tag (TMT) labeling reagents. Upon isolation and subsequent fragmentation of each peptide, reporter ions corresponding to each TMT reagent provide relative abundances for those peptides in each sample. Thermo Scientific Proteome Discoverer software uses this data to calculate the relative amount of total protein in each sample, which is used to normalize the data post analysis via liquid chromatography-tandem mass spectrometry

Immunoprecipitated exosomes were eluted from the Dynabeads at 95 °C for 5 min in lysis buffer (100 μL, 12 mM sodium lauroyl sarcosine, 0.5% sodium deoxycholate, 50 mM triethylammonium bicarbonate (TEAB), Halt™ Protease, and Phosphatase Inhibitor Cocktail), then subjected to bath sonication (10 min, Bioruptor Pico, Diagenode Inc.; Denville, NJ, USA). The samples were treated with tris (2-carboxyethyl) phosphine (10 μL, 55 mM in 50 mM TEAB, 30 min, 37 °C), followed by treatment with chloroacetamide (10 μL, 120 mM in 50 mM TEAB, 30 min, 25 °C in the dark). They were then diluted fivefold with aqueous 50 mM TEAB and incubated overnight with Sequencing Grade Modified Trypsin (1 μg in 10 μL of 50 mM TEAB; Promega, Cat # V511A, Madison, WI, USA). Following this, an equal volume of ethyl acetate/trifluoroacetic acid (TFA, 100/1, *v*/*v*) was added, and after vigorous mixing (5 min) and centrifugation (13,000× *g*, 5 min), the supernatants were discarded, and the lower phases were dried in a centrifugal vacuum concentrator. The samples were then desalted using a modified version of Rappsilber’s protocol [45], in which the dried samples were reconstituted in acetonitrile/water/TFA (solvent A, 100 μL, 2/98/0.1, *v*/*v*/*v*) and then loaded onto a small portion of a C18-silica disk (3M, Maplewood, MN, USA) placed in a 200 μL pipette tip. Prior to sample loading, the C18 disk was prepared by sequential treatment with methanol (20 μL), acetonitrile/water/TFA (solvent B, 20 μL, 80/20/0.1, *v*/*v*/*v*), and finally with solvent A (20 μL). After loading the sample, the disc was washed with solvent A (20 μL, eluent discarded) and eluted with solvent B (40 μL). The collected eluent was dried in a centrifugal vacuum concentrator. The samples were then chemically modified using a TMT11plex Isobaric Label Reagent Set (Thermo Fisher Scientific, Cat # A34808, Waltham, MA, USA) as per the manufacturer’s protocol. The TMT-labeled peptides were dried and reconstituted in solvent A (50 μL), and an aliquot (2 μL) was taken for measurement of total peptide concentration (Pierce Quantitative Colorimetric Peptide, Thermo Fisher Scientific, Waltham, MA, USA). The samples were then pooled and desalted again using the modified Rappsilber’s protocol. The eluants were then dried and reconstituted in water/acetonitrile/FA (solvent B, 10 μL, 98/2/0.1, *v*/*v*/*v*), and aliquots (5 μL) were injected onto a reverse-phase nanobore HPLC column (AcuTech Scientific, C18, 1.8 μm particle size, 360 μm × 20 cm, 150 μm ID, San Diego, CA, USA), equilibrated in solvent E, and eluted (500 nL/min) with an increasing concentration of solvent F (acetonitrile/water/FA, 98/2/0.1, *v*/*v*/*v*: min/% F; 0/0, 5/3, 18/7, 74/12, 144/24, 153/27, 162/40, 164/80, 174/80, 176/0, 180/0) using an Eksigent NanoLC-2D system (Sciex, Framingham, MA, USA)). The effluent from the column was directed to a nanospray ionization source connected to a hybrid quadrupole-Orbitrap mass spectrometer (Q Exactive Plus, Thermo Fisher Scientific, Waltham, MA, USA), acquiring mass spectra in a data-dependent mode alternating between a full scan (350–1700 m/z, automated gain control (AGC) target 3 × 10^6^, 50 ms maximum injection time, FWHM resolution 70,000 at 200 m/z) and up to 15 MS/MS scans (quadrupole isolation of charge states 2–7, isolation window 0.7 m/z) with previously optimized fragmentation conditions (normalized collision energy of 32, dynamic exclusion of 30 s, AGC target 1 × 10^5^, 100 ms maximum injection time, FWHM resolution 35,000 at 200 m/z). 

### 7.7. Proteomics Analysis

Raw proteomic data were searched against the Uniprot human-reviewed protein database using SEQUEST-HT in Proteome Discoverer (Version 2.4, Thermo Scientific, Waltham, MA, USA), which provided measurements of abundances for identified peptides in each sample that were normalized to total protein amount. Decoy database searching was used to identify high confidence tryptic peptides (FDR < 1%). Tryptic peptides containing amino acid sequences unique to individual proteins were used to identify and provide relative quantification between proteins in each sample. Normalized protein abundances for pre- and post-work samples from each participant were scaled so that the average abundance was 100. Median abundance values of all replicates from each condition were used to generate abundance ratios for each protein (post-work/pre-work). 

### 7.8. Protein Bioinformatics Analysis

Proteins exhibiting a fold change with a magnitude ≥ 1.2 and a *p* value ≤ 0.1 were subject to comprehensive gene-set enrichment-analysis gene ontology (GO) classification and KEGG [24] pathway analysis using Enrichr (Chen et al., 2013), as well as functional protein association network analysis using the STRING database (version 11.5), which was used for functional interpretation of the proteomics data and provided *p*-values corrected by the FDR method [25]. The relatively high *p* value of ≤ 0.1 was set due to the limited number of samples that were available for analysis; the selection of a higher significance threshold allowed us to expand the number of proteins to be assessed as potential biomarkers of fatigue in the Validation group so that potential biomarkers would not be excluded. We note that this significance value was only used for the selection of proteins for the additional analyses reported above and not for the final presentation of findings of potential biomarkers.

### 7.9. Targeted LC-MS/MS Protein Quantification

Proteins isolated by antibody-conjugated microbeads were reduced, alkylated, and treated with trypsin as described in *Global Proteomics Analysis*; however, in contrast with that sample processing protocol, no isotopically labeled chemical tags were utilized to provide relative quantification between peptides in different samples. Furthermore, the data were acquired with the mass spectrometer utilizing a customized target-selected ion monitoring/data-dependent MS/MS (t-SIM/dd-MS^2^) method in which an inclusion list was used to isolate and fragment select peptides corresponding to specific proteins and measure precursor ion peak areas. Data from the global proteomic analysis were used to identify unique peptides for this analysis and select the correct m/z (Da) and charge state (Z) of each peptide targeted. The sensitivity gained by the targeted analysis using the SIM scan (AGC target 2 × 10^5^, 130 ms maximum injection time, FWHM resolution 70,000 at m/z 200, isolation window 2.0 m/z) permitted modification of the LC gradient (min/% F; 0/0, 5/3, 55/22, 61/35, 63/80, 73/80, 75/0, 79/0) and shortening of mass spectrometer acquisition time. 

### 7.10. Global miRNA Analysis

RNA was extracted from the immunoprecipitated salivary exosomes using the SeraMir Exosome RNA Column Purification Kit (System Biosciences, Palo Alto, CA, USA) according to the manufacturer’s protocol. The quality of the RNA was assessed via electrophoresis using the Small RNA Kit (Agilent Technologies, Santa Clara, CA, USA) on a 2100 Bioanalyzer System (Agilent Technologies) according to the manufacturer’s instructions. Global profiling of miRNA from Test and Discovery group samples was completed at the UCLA Center for Systems Biomedicine with the nCounter Human v3 miRNA Expression Assay (NanoString Technologies; Seattle, WA, USA), in which 800 pairs of probes specific for a predefined set of biologically relevant miRNAs were combined with a series of internal controls to form a Human miRNA Panel CodeSet (NanoString Technologies, Seattle, WA, USA). miRNA (100 ng) targets of interest were hybridized overnight with two juxta-positioned probes: a biotinylated capture probe and a uniquely fluorescently labeled reporter probe for each target. The hybridized samples were then transferred to the nCounter Prep Station, where excess probes were removed, and the target–probe complexes were immobilized and aligned on the surface of a flow cell using an automated liquid handler. The unique sequences of the reporter probes were counted using the nCounter Digital Analyzer and translated into the number of counts per miRNA target. nSolver Analysis Software (NanoString Technologies, Seattle, WA, USA) was used to facilitate data extraction and analysis. In Nanostring analysis, the raw miRNA counts were normalized (corrected for multiple testing) using positive control (spike-in) normalization and the geometric mean of the top 100 expressed miRNAs, performed using the Benjamini–Hochberg method within the Nanostring nSolver software.

### 7.11. Targeted miRNA Analysis 

Quantitative polymerase chain reaction (qPCR) was used to verify the miRNA levels detected in the NanoString analysis. For verification, the same RNA samples that were used in the NanoString analysis were assayed for select miRNAs using the TaqMan Advanced miRNA Assay (Thermo Fisher Scientific, Cat # A25576, Waltham, MA, USA) according to the manufacturer’s protocol. qPCR amplification Ct values for each resident’s pre- and post-shift samples were compared using the ΔΔCt method and converted to a % change in miRNA levels (post-work—pre-work) for each subject. The RNA yield from exosomes isolated from 1 mL of saliva was, on average, 208 +/− 44 ng. This yield is comparable to published reports of saliva exosome RNA yields, specifically 209–274 ng/mL of salivary exosomes [46].

### 7.12. Identification of miRNA Target Genes

Target genes associated with miRNAs exhibiting significant changes in abundance in response to work shifts were identified using miRNet [47]. Proteins corresponding to these genes were subsequently checked for and identified in the list of proteins identified in the global proteomics analysis. Potential miRNA target genes were identified when the direction of change in the abundance of a miRNA was opposite that of a protein encoded by its regulated gene.

### 7.13. Statistical Analysis

PoMS analysis. Both pre- and post-shift PoMS scores from all 36 residents were included in the analysis. Wilcoxon signed-rank test was used for the paired comparison between pre- and post-work shifts. The analysis was performed using SAS version 9.4 (SAS Institute Inc., Cary, NC, USA).

Proteomics analysis. A Student’s *t*-test was used to determine if the observed differences in protein abundances between pre- vs. post-work for each group were statistically significant (*p* < 0.05). The *p* value of ≤0.1 described above was solely used to select proteins for further analysis and assessment in the Validation group and not for the identification of potential biomarkers of fatigue.

miRNA analysis. A paired *t*-test was used to identify miRNAs exhibiting significant changes in abundance in response to work shifts (fold change magnitude ≥ 1.2; *p* value < 0.05). For identifying miRNA associated with PoMS scales in the discovery group, median values of FI and TMD pre- and post-work were calculated, and candidate miRNA biomarkers were determined using the Wilcoxon ranked-sum test with a false discovery rate (q) of <0.05.

### 7.14. Study Design

The study design and flow scheme, including enrollment, saliva collection, and PoMS assessment pre- and post-work shift, calculation of TMD and subscales scores for separation into Test, Discovery, and Validation groups, and exosome isolation and analyses are shown in Figure 1.

## Figures and Tables

**Figure 1 ijms-23-05257-f001:**
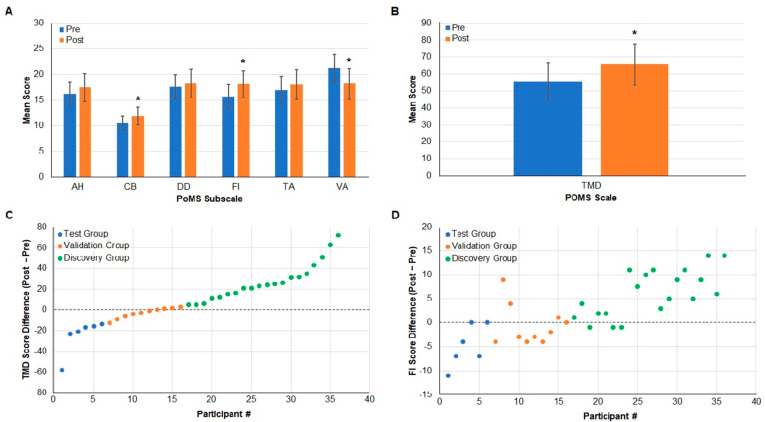
Difference in Total Mood Disturbance (TMD) as assessed by the Profile of Mood States (PoMS) allows separation into Discovery, Validation, and Test groups. (**A**) The PoMS scores for anger–hostility (AH), confusion–bewilderment (CB), depression–dejection (DD), fatigue–inertia (FI), tension–anxiety (TA), and vigor–activity (VA) are shown pre- and post-work shift. CB * *p* = 0.003, FI * *p* = 0.0393, and VA * *p* = 0.0047. (**B**) The combined TMD score for all participants’ pre- and post-work shifts is shown; * *p* = 0.0200. (**C**) The number of participants sorted into each group (*x*-axis), and the difference between pre- and post-work shift TMD (*y*-axis) is shown. The 6 participants with the greatest decrease in TMD score are in the Test group (blue), the 20 participants with the greatest increase in the Discovery group (green), and 10 intermediate participants in the Validation group (orange). (**D**) The number of participants sorted into each group (*x*-axis), and the difference between pre- and post-work shift FI (*y*-axis) is shown.

**Figure 2 ijms-23-05257-f002:**
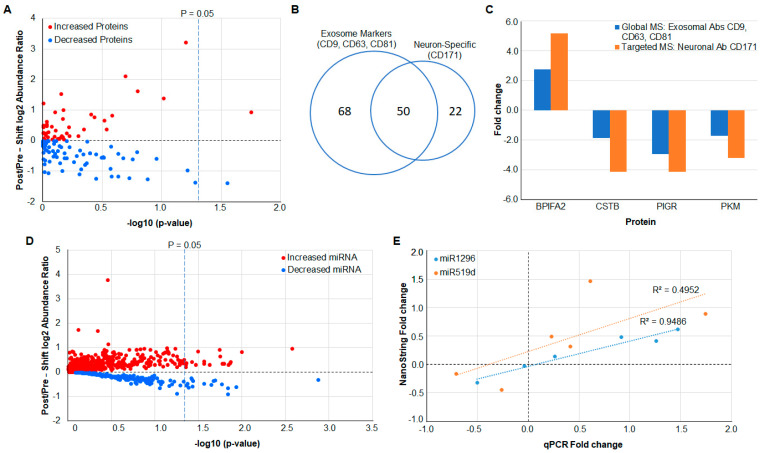
Multiomics analysis reveals quantifiable differences in protein and miRNA abundances in the Test group pan- and neuron-derived exosomes. (**A**) Differences in the abundances of proteins present in the salivary exosomes of Test group participants pre- and post-work shifts are illustrated via a volcano plot. The log_10_ (abundance ratio: post-work/pre-work) is plotted against −log_10_ (*p* value). (**B**) A Venn diagram shows the overlap of proteins found in exosomes isolated with pan-exosomal or neuron-selective exosomal markers for a single participant. (**C**) The protein fold change for the single Test group participant is shown when using global proteomics on exosomes isolated by a pan-exosome set of antibodies (Global MS (exosome abs: CD9, CD63, CD81); blue bars) or targeted MS on four corresponding to the four proteins (STVSSLLQK/BPIFA2, SQVVAGTNYFIK/CSTB, TVTINCPFK/PIGR, LDIDSPPITAR/PKM) present in exosomes isolated using an antibody to a neuron-specific exosome marker (Targeted MS (neuronal ab: CD171); orange bars). (**D**) Differences in the abundances of miRNAs present in the salivary exosomes of participants pre- and post-work shift are illustrated via a volcano plot. The log_10_ (abundance ratio: post-work/pre-work) is plotted against −log_10_ (*p*-value). (**E**) Fold change of miR-519d and miR-1296 as measured using the NanoString platform (*y*-axis) shows a moderate and strong positive correlation (miR-519d R^2^ = 0.50, r (5) = 0.70, *p* = 0.01881; miR-1296 R^2^ = 0.95, r (5) = 0.97, *p* = 0.0062) with fold change as measured by qPCR (*x*-axis).

**Figure 3 ijms-23-05257-f003:**
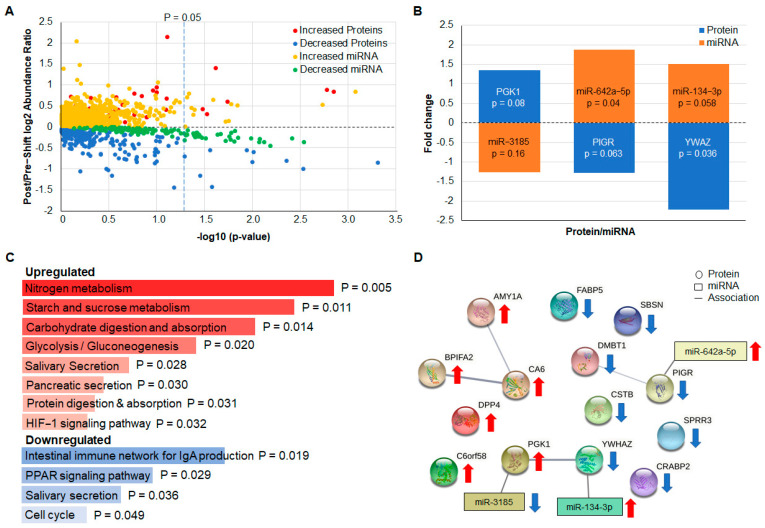
Interconnected protein and miRNA networks regulate molecular pathways associated with increased TMD score in the Discovery group. (**A**) Differences in the abundances of proteins and miRNAs present in the salivary exosomes of Discovery group participants (*n* = 20) pre- and post-work shift are illustrated via volcano plot; significant differences are points to the right of the vertical line marked *p* = 0.05. The log_10_ (abundance ratio: post-work/pre-work) is plotted against the −log_10_ (*p*-value). (**B**) The mean fold change of three significantly altered miRNA (miR-3185, miR-642-5p, miR-134-3p) is shown to inversely relate to the abundances of proteins encoded by one of their target genes (PGK1, PIGR, YWHAZ). (**C**) Gene set enrichment analysis of upregulated and downregulated proteins using Enrichr [23] shows enrichment of KEGG database [24] molecular pathways after a 12 h work shift. Results are ranked according to *p* value. (**D**) Functional protein association network analysis using STRING [25] shows that associated and interacting protein networks are differentially regulated after a 12 h work shift.

**Figure 4 ijms-23-05257-f004:**
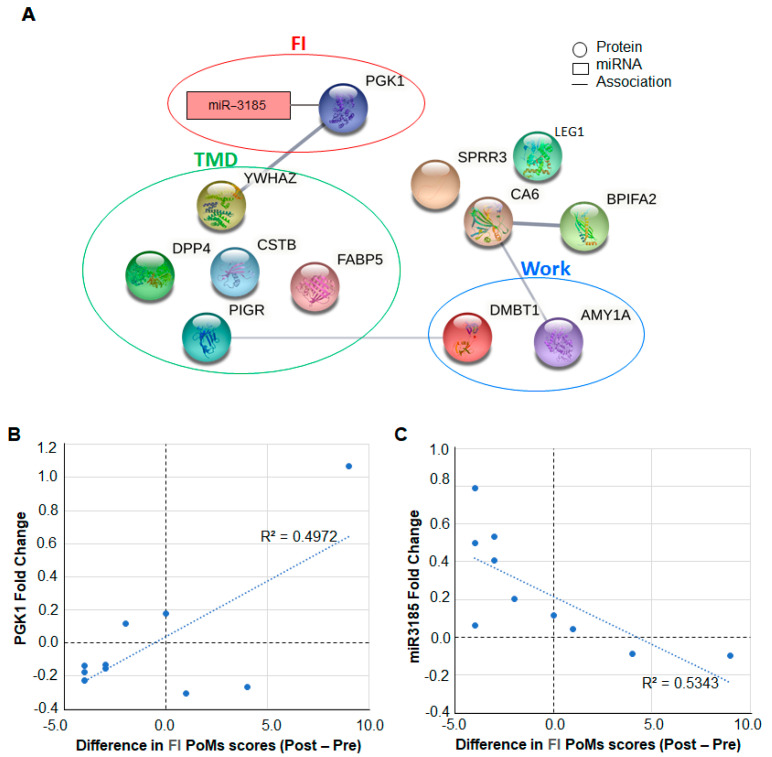
Proteins and miRNA identified in the Validation group are associated with Total Mood Disturbance (TMD), Fatigue–Inertia (FI), and Work, and (**A**) 12 of the 14 proteins identified as significantly altered in the Discovery group were also identified in the Validation group. The abundance of one protein (PGK1) corelates with FI (red circle), and those of the five proteins (DPP4, PIGR, CSTB, FABP5, and YWHAZ) are associated with TMD (green circle). Of the remaining six proteins, two (AMY1A and DMBT1) were altered significantly (*p* < 0.05) after a 12 h work shift (blue circle), while the remaining four (BPIFA2, CA6, SPRR3, and LEG1) were not altered. (**B**) PGK1 fold change (*y*-axis) shows a weak positive correlation (R^2^ = 0.50, r (10) = 0.70, *p* = 0.0242) and (**C**) miR3185 fold change (*y*-axis) shows a weak negative correlation (R^2^ = 0.53, r (10) = −0.73, *p* = 0.0165) with FI difference in Validation group participants after a work shift.

**Figure 5 ijms-23-05257-f005:**
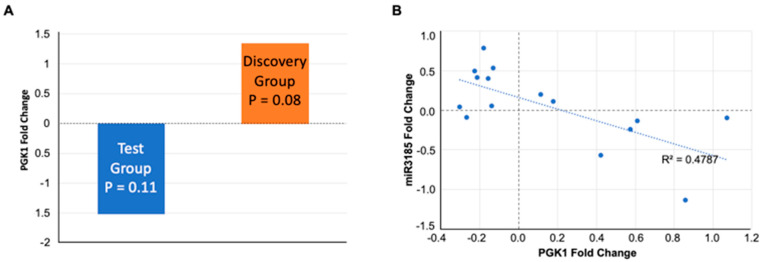
PGK1 levels in Test and Discovery groups, and correlation of PGK1 to mi3185. (**A**) The PGK1 fold change for the Test and Discovery groups is shown. (**B**) The relative PGK1 (*x*-axis) and miR3185 levels (*y*-axis) display an inverse relationship. The correlation coefficient is R^2^ = 0.48 r (15) = 0.69, *p* = 0.0044).

**Figure 6 ijms-23-05257-f006:**
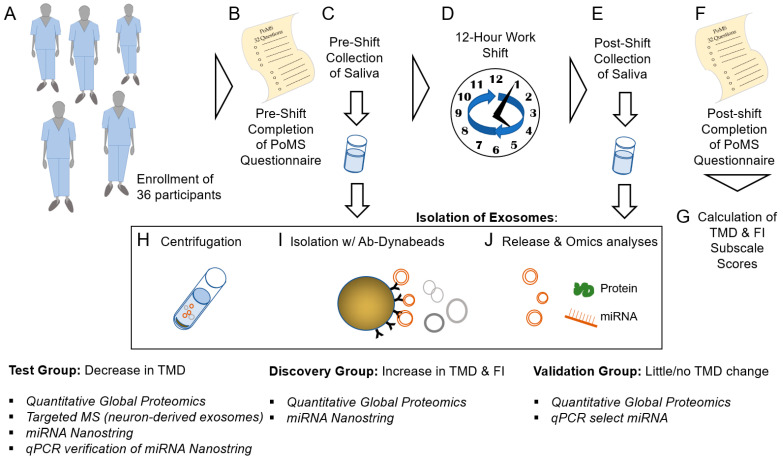
Study flow scheme for Profile of Mood States (PoMS) assessment, sample collection, processing, and analysis. (**A**) Resident participants (*n* = 36) were enrolled. (**B**) Participants completed the PoMS questionnaire and (**C**) collected saliva before a (**D**) 12 h work shift. (**E**) Saliva was collected, and the (**F**) PoMS questionnaire completed again after the work shift. (**G**) Total Mood Disturbance (TMD) and subscale (for example, fatigue–inertia, FI) scores were calculated. Isolation of exosomes comprised (**H**) centrifugation, (**I**) binding of exosomes to exosome marker-specific (and in one instance, neuronal-marker specific) antibody-conjugated Dynabeads, (**J**) release of exosomes for processing, and omics analyses. Based on TMD, participant samples were separated into Test (decrease in TMD or ‘improved’ mood), Discovery (increased in TMD or mood disturbance), and Validation (little/no change in TMD) groups. Exosomes from each group underwent the analyses shown.

**Table 1 ijms-23-05257-t001:** Identification of differentially abundant proteins and miRNA in salivary exosomes in the Discovery group. The protein–miRNA pairs are listed in the same row.

Gene Symbol	Protein Description	Protein Fold Change: Post-/Pre-Work	Protein *p*-Value: Post-/Pre-Work	miRNA	miRNA Fold Change: Post-/Pre-Work	miRNA *p*-Value: Post-/Pre-Work
LEG1	Liver-enriched gene 1 protein	4.46	0.079			
AMY1A	Alpha-amylase 1 ^n,c^	1.85	0.002			
BPIFA2	BPI fold-containing family A member ^n^	1.64	0.006			
CA6	Carbonic anhydrase 6 ^n^	1.5	0.083			
PGK1	Phosphoglycerate kinase 1 ^n^	1.34	0.08	hsa-miR-3185	0.79	0.016
DPP4	Dipeptidyl peptidase 4	1.24	0.03			
SBSN	Suprabasin	−1.23	0.06			
PIGR	Polymeric immunoglobulin receptor ^m,n,c^	−1.28	0.063	hsa-miR-642a-5p	1.9	0.04
CSTB	Cystatin-B ^n,c^	−1.34	0.057			
SPRR3	Small proline-rich protein 3	−1.49	0.01			
CRABP2	Cellular retinoic acid-binding protein 2	−1.49	0.098			
FABP5	Fatty acid-binding protein 5	−1.72	0.004			
YWHAZ	14-3-3 protein zeta/delta ^n,c^	−2.22	0.036	hsa-miR-134-3p	1.5	0.058
DMBT1	Deleted in malignant brain tumors 1 ^n,c^	−2.63	0.027			

**^m^** = Membrane Protein; **^n^** = Experimentally identified using neuron-specific marker (CD171); **^c^** = Relevant in Chronic Fatigue Syndrome.

## Data Availability

Not applicable.

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
