# Peer review of "Integrated Multiomics Analysis of Salivary Exosomes to Identify Biomarkers Associated with Changes in Mood States and Fatigue"

_ijms, 2022, doi:10.3390/ijms23095257_

Round 1

Reviewer 1 Report

This is an original work analysing proteins and miRNA from salivery exosomes associated with mood changes. The title is too specific however as many of the results are correlations with mood state and not fatigue alone. More highlighted segration of the discussion of these results between TMD score and fartigue will enhance the paper further. Furthermore, please discuss the influence of fatigue on TMD.

Please discuss how much protein and miRNA the exosomes yielded, do the amounts correspond to previous studies?  How was the the amount of protein and miRNA normalised for all samples prior to analyis.

The use of the neuron-derived exosomes to validate the protein measurements is confusing. Was this carried out for all groups? If not please highlight this in all aspects of the paper and discuss these results separately from the global analysis approach.

Please address why such a high p-value (<0.1 ) was used in the protein bioinformatic analysis.

Please add in the raw qPCR results into the supplementary data. Currently there are only correlations and so the inital qPCR data cannot be critiqued.

Another limitation should include that the predicted miRNA protein targets require validation.

Author Response

Please response attached.

Reviewer 2 Report

In overall, the paper has a potential to be an interesting piece for future readers. However, in the current form it needs to be re-worked.

After reading this paper I have an impression that:

The experiment model was based on saliva examination before and after work shift solely.

What You have done is that You added a very specific narrative, that is rather not justified by the model itself:

  1. You are mentioning cognitive fatigue, what was rather not examined in the objective manner (by measuring cognitive function in repeated manner). Are You referring to fatigue itself? Why did You not use any kind of fatigue-specific questionnaires?
  2. If point 1 is true, then I think that it would be better to not describe any casuse-efect relationship between (cognitive?) fatigue and biomarkers. What You have done is You have examined biomarkers level before and after shift, and You can correlate that with some domains of a questionnaire results, is that correct? Correlation does not imply causation here.
  3. The section that I am completely lost in the section “Separation of saliva samples by PoMS TMD score”. Why did You used different methods in those subgroups? What was the reason to create those subgroups after all? I think that You should describe results from those subgroups in separate papers. Otherwise I am afraid that the conclusions from the above paper (based on different groups examined using different methods) are not justified and might be misleading for readers.

Abstract „

“objectively assess CF would be of value” In my humble opinion, fatigue is a kind of qualia, therefore what we can do is to examine correlates of such subjective experience. In terms of cognitive fatigue, by definition, it could be objectively assessed using cognitive function tests. What You did in Your study is to examine the correlate (a biochemical factor) of cognitive fatigue (confirmed objectively using cognitive function tests).

“The objective of the study described herein was identification of molecular signals associated with fa-tigue that may lead to CF in saliva-borne, brain-derived exosomes” Are You sure that Your model of experiment that You have applied allows on drawing conclusions on cause and effects?

“salivary exosomes (…)  indicative of changes in mood states that may affect cognitive performance” In this sentence You draw a very complicated conclusion based on cause and effect relationship between three variables. Are You sure that You can draw such conclusions from Your own experiment?

Introduction:

“Fatigue-associated changes in overall physical state such as the frequently reported dryness of the mouth” can You add a citation to this?

“We hypothesize that the potential for salivary exosomes to reflect changes in cogni-tive function” What do You think about a potential relationship between autonomic nervous system function with saliva production control and cognitive function? Presumably, autonomic nervous system might be related to both of those factors.

Materials and methods

“Saliva samples were collected over a period of 60 minutes immediately before and following a 12-hour work shift in 50 mL conical tubes” as I suppose, what was not controlled is the pattern of behaviour throughout those 12 hours (physical activity level, stress level, presence of naps (?)). Moreover, such experimental model would not take into account potential effects of circadian rhythm on examined variables. If You agree that that was the case, then please add this into discussion as potential limitations of the study. Do all of these shifts took part solely during day r night? If not, then it would be an additional limitation of the study.

This section lacks of description of cognitive fatigue measurement and statistical analysis.

Results

“Select proteins and miRNAs identified from the larger Discovery group analysis were measured and any correlation to PoMS subscales were determined.” I am lost with this sentence.

“Statistics per-formed using a Wilcoxon Signed Rank Test where CB *p = 0.003, FI *p = 0.0393, and VA *p = 0.0047.” information on tests applied should be in statistical analysis section, results should be in the main text, not in the figure caption. What about calculation of the effect size for each comparison? What about resolving problems caused by multiple comparisons?

“significantly changed in between pre- and post-work shift (absolute fold change ≥ 1.2, p ≤ 0.05)” of the alpha level was set on 0.05, then values lower than 0.05 should be considered as significant. AS I have already mentioned, because of the lack of statistical methods description ,result section is not clear to me.

Table 1 is not fully visible

Supplementary Fig. S2 – why one model is non-linear? What kind of models were applied in overall?

Discussion seems to be rather nice. However, in the current form I do not know if it is referring to validate conclusions from Your study.  

Author Response

Please see responses attached.

Round 2

Reviewer 2 Report

I do appreciate the extensive point-by-point answers of Authors and fixes implemented. In general, I suppose that the aim of the paper is significant in lives of majority of people around the world, and finding a biomarker of fatigue that is a relatively easy to examine might potentially vastly improve it. However, I think that still, two major (points no 2 and 3) and two minor (points no 1 and 4) issues resolves.

  1. In the Introduction to Abstract and even more in the main text You do define cognitive fatigue on basis of deterioration of cognitive function measured in a n objective manner while You have no such indicator in Your study. It would be better to explain cognitive fatigue bases on the previous studies that have measured it on the basis of subjective method (for instance using POMS or Chalder Fatigue Scale with mental sub-score) and correlated this with objective results. It would explain why Your subjective method might be also valuable in cognitive fatigue assessment.
  2. Regarding statistical analysis. It would be much better to describe all methods applied in one subsection (typically, it is the last from materials and methods).
    • You have added that “The relatively high p-value of ≤ 0.1 was set due to the limited of number of samples that were available for analysis” I have no problem with that, but what about the Academic Editor? Does the conventional (0.05) alpha level could be omitted here?
    • Then in the next paragraph You have added “performed using the Benjamini-Hotchberg method within the Nanostring nSolver software”. Are You referring to the “Benjamini-Hochberg”?
    • Still, You have not calculated effect sizes for comparisons, therefore You can list it as an additional study limitation
    • In the paper and in the response You have used the notation “p-value ≤ 0.05” which I would recommend to change. The standard alpha value in biomedical science is „0.05”. It is a threshold. P-value below this threshold could be regarded as statistically significant (in this awful ritual that is the main base in statistical analysis process).
    • “AsIn the single Test group participant from which neuron-derived exosomes were isolated, as” If that is the case, then this should be underlined in the discussion and abstract.
  3. Point no 3 refers to the study model itself. Why did You chose a subgroup of patients with different direction of changes in questionnaire (towards improvement after night shift) as a “test group”? In my opinion, by doing so You have potentially lost a lot of important information: what if the same potential biomarkers of fatigue would change in different direction in this subgroup vs those that experienced fatigue? And why some analysis were done one patient only (point no 2.5)? In my opinion, it means that the research model has been poorly constructed, or that the above paper is describing a preliminary results or is a purely explorative analysis. If the first is true, then I would recommend to reject this paper, if the latter is true, then I would leave the decision to Academic Editor if the paper should be published in J. Mol. Sci.
  4. In results You have added “As described in Methods, TMD score was subsequently used to segregate saliva sample into 3 groups: Test, Discovery and Validation (Fig. 2C). This is similar to a previously reported approach used in analysis of saliva samples for biomarkers of traumatic brain injury (Pietro et al., 2018).” I am not sure if references could be added to results section, Academic Editor has to decide here.

Author Response

Reviewer #2

General comment from reviewer on first round responses: I do appreciate the extensive point-by-point answers of Authors and fixes implemented. In general, I suppose that the aim of the paper is significant in lives of majority of people around the world, and finding a biomarker of fatigue that is a relatively easy to examine might potentially vastly improve it. However, I think that still, two major (points no 2 and 3) and two minor (points no 1 and 4) issues resolves.

Author Response: We thank the reviewer for the comment. Yes, we agree with the reviewer that the findings of a biomarker of fatigue that is relatively easy to assess from saliva using a rapid test would benefit the lives of majority of people around the world. Our study of saliva exosomes from residents after a 12-hour shift is a first step in developing such a test. The comments for points 1-4 are addressed below.

Comment #1: In the Introduction to Abstract and even more in the main text you do define cognitive fatigue on basis of deterioration of cognitive function measured in an objective manner while you have no such indicator in your study. It would be better to explain cognitive fatigue based on the previous studies that have measured it on the basis of subjective method (for instance using POMS or Chalder Fatigue Scale with mental sub-score) and correlate this with objective results. It would explain why your subjective method might be also valuable in cognitive fatigue assessment.

Author Response: We thank the reviewer for the comment. One of the key references we present is Fogt et al., 2010, wherein the authors report fatigue as assessed by PoMS directly correlated with decreased cognitive performance determined by the Stroop Color-Word Conflict Test. In the revised the manuscript we specifically state the finding from Fogt et al., this is in the second-to-last paragraph of the Introduction section, and the correlation to cognitive fatigue and performance is mentioned again in the Methods section, this is in the paragraph on Profile of of Mood states. As well we mention it in the paragraph on limitations of our study in Discussion section of the revised manuscript.

Comment #2.: Regarding statistical analysis. It would be much better to describe all methods applied in one subsection (typically, it is the last from materials and methods).

Author Response: We thank the reviewer for the comment.  All statistical methods are now together in one subsection near the end of Methods (but just before Study Design), under ‘Statistical considerations’.

Comment #2.1: You have added that “The relatively high p-value of ≤ 0.1 was set due to the limited of number of samples that were available for analysis” I have no problem with that, but what about the Academic Editor? Does the conventional (0.05) alpha level could be omitted here?

Author Response: We have clarified under ‘Protein bioinformatic analysis’ and ‘Statistical considerations’ in Methods section of the revised manuscript that the p-value of ≤ .1 was only used as a screen for selection of proteins to go onto further analysis and not for any final presented results on potential biomarkers of fatigue.

Comment #2.2: Then in the next paragraph you have added “performed using the Benjamini-Hotchberg method within the Nanostring nSolver software”. Are You referring to the “Benjamini- Hochberg”?

Author Response: We apologize for the spelling error. Yes, it is Benjamini-Hochberg and the typo has been corrected in the revised manuscript.

Comment #2.3: Still, you have not calculated effect sizes for comparisons, therefore you can list it as an additional study limitation.

Author Response: We have now added a statement on effect sizes per the reviewer comment in the Discussion section of the revised manuscript.

Comment #2.4: In the paper and in the response, you have used the notation “p-value ≤ 0.05” which I would recommend to change. The standard alpha value in biomedical science is „0.05”. It is a threshold. P-value below this threshold could be regarded as statistically significant (in this awful ritual that is the main base in statistical analysis process).

Author Response: We assume the reviewer was requesting removal of ‘0’ before the decimal and capitalizing ‘P’ of P-value; we have made those changes in the revised manuscript.  We apologize if and this is not what the reviewer meant and we didn’t fully understand the comment.

Comment #2.5: “As in the single Test group participant from which neuron-derived exosomes were isolated, as” If that is the case, then this should be underlined in the discussion and abstract.

Author Response: Since the neuron-derived exosomes were from a single individual, the results for that study are not specifically mentioned in the Abstract. We do mention it in the Discussion that the neuron-derived exosomes from a single participant was used to qualitatively compare protein abundances obtained to that from the pan-exosome isolated samples.  We also point out in the study limitation statement, that the neuron-derived exosomes were only isolated from a single participant sample.

Comment #3: Point no 3 refers to the study model itself. Why did you choose a subgroup of patients with different direction of changes in questionnaire (towards improvement after night shift) as a “test group”? In my opinion, by doing so you have potentially lost a lot of important information: what if the same potential biomarkers of fatigue would change in different direction in this subgroup vs those that experienced fatigue? And why some analysis were done one patient only (point no 2.5)? In my opinion, it means that the research model has been poorly constructed, or that the above paper is describing a preliminary results or is a purely explorative analysis. If the first is true, then I would recommend to reject this paper, if the latter is true, then I would leave the decision to Academic Editor if the paper should be published in J. Mol. Sci.

Author Response:  We apologize for the confusion resulting from how we described the roles of the three groups. As outlined in the Results section, the Test group samples were used to establish the validity of the analytical methods and for testing of potential biomarkers. We have now added this sentence in Methods section of the revised manuscript under ‘Separation of saliva samples by PoMS TMD score’:

 “The three groups were established based on the hypothesis that potential biomarkers of fatigue ‘discovered’ in individuals reporting fatigue (Discovery group) could be tested for their potential as biomarkers in the group that reported an opposite ‘improved’ change in mood (Test group) based on the supposition that these biomarkers would either be unchanged or change in the opposite direction in the Test group. The potential biomarkers were again ‘validated’ in the group with a mix of scores (Validation group).” 

Both the Discovery and Test groups samples underwent global proteomics analysis, and no information was lost.

In Results section under ‘PGK1 protein and miR3185 in saliva as potential biomarkers of fatigue’, we indicate in the manuscript that for this promising biomarker candidate, “In Test group participants salivary exosomes (with a lower rather than increased TMD score post-shift), the fold change in PGK1 was negative and in the Discovery group participants who recorded increased TMD and FI scores, it was positive (Fig. 6A).”

On the reviewer comment that some analysis was done only on one (patient):  As we state in response to reviewer comment #2.5, the main findings described in the paper, and the only findings presented in the Abstract, were based on analyses from all 36 residents in the study. The findings from neuron-derived exosomes from one individual, we believe is both of interest and important for the study in confirming qualitatively that exosomes of neuronal origin are a major component of saliva exosomes. We have endeavored to present this part in the manuscript as only a sub-study of the overall study.

We humbly disagree with the reviewer that the model we use for segregation of the saliva samples, which is similar to a previously reported study (Di Pietro et al.,2018), has been poorly constructed or the manuscript is describing only preliminary data or is purely explorative in nature.

Comment #4: In results You have added “As described in Methods, TMD score was subsequently used to segregate saliva sample into 3 groups: Test, Discovery and Validation (Fig. 2C). This is similar to a previously reported approach used in analysis of saliva samples for biomarkers of traumatic brain injury (Pietro et al., 2018).” I am not sure if references could be added to results section, Academic Editor has to decide here.

Author Response: We thank the reviewer for noting this and the Di Pietro reference for the method was removed from Results and is now in Methods under ‘Separation of saliva samples by PoMS TMD score’.

Round 3

Reviewer 2 Report

Author Response on comment 2.4: “We assume the reviewer was requesting removal of ‘0’ before the decimal and capitalizing ‘P’ of P-value; we have made those changes in the revised manuscript.  We apologize if and this is not what the reviewer meant and we didn’t fully understand the comment.”

  1. Indeed, it was not my intention. I just simply implied that notation “p-value ≤ 0.05” is wrong in the currently established procedure in statistical analysis in biomedicine field. In fact, if alpha = 0.05 then p-value less than 0.05 would be „significant”. Therefore, You should write is as “p-value < 0.05” no as “p-value ≤ 0.05”. Please change it accordingly.

I would also like to add following comments:

  1. Thank You for the clarification of experimental model made in the current form of manuscript. I believe that it would be more clear for future readers now. However, I would still insist on the fact, that instead of “For identifying miRNA associated with PoMS scales in the discovery group, median 343 values of FI and TMD pre and post work were calculated, and candidate miRNA biomarkers were determined using the Wilcoxon ranked-sum test” it would be much better to correlate changes of results in scale that indicate change in fatigue after night shift with changes in candidate miRNA biomarkers.
  2. Moreover, I would add that it was an explorative study into limitations in discussion, as no specific hypothesis were established before the conduction of the study? Or have You been testing a specific set of pre-determined hypothesis using the Wilcoxon ranked-sum test in this case? I believe that it is nothing wrong for the study to be explorative in nature.
  3. Subsection title “Statistical considerations” should be changes to „Statistical analysis”. In a standard arrangement, it is the last part of materials and methods section. Therefore, I would move subsection „Study Design” somewhere else in materials and methods section.

Looking on comment my comment no 2, I would like the decision on publication of this paper to the Academic Editor.